# The Role of Histo-Blood Group Antigens and Microbiota in Human Norovirus Replication in Zebrafish Larvae

Arno Cuvry,[a] Roberto Gozalbo-Rovira,[b] Dufie Strubbe,[a] Johan Neyts,[a] Peter de Witte,[c] Jésus Rodríguez-Díaz,[b] Joana Rocha-Pereira[a]

[a]KU Leuven-Department of Microbiology, Immunology and Transplantation, Rega Institute, Laboratory of Virology and Chemotherapy, Leuven, Belgium
[b]Department of Microbiology, School of Medicine, University of Valencia, Valencia, Spain
[c]KU Leuven-Department of Pharmaceutical and Pharmacological Sciences, Laboratory for Molecular Biodiscovery, Leuven, Belgium

Roberto Gozalbo-Rovira and Dufie Strubbe contributed equally to this work.

**ABSTRACT** Human norovirus (HuNoV) is the major agent for viral gastroenteritis, causing >700 million infections yearly. Fucose-containing carbohydrates named histo-blood group antigens (HBGAs) are known (co)receptors for HuNoV. Moreover, bacteria of the gut microbiota expressing HBGA-like structures have shown an enhancing effect on HuNoV replication in an *in vitro* model. Here, we studied the role of HBGAs and the host microbiota during HuNoV infection in zebrafish larvae. Using whole-mount immunohistochemistry, we visualized the fucose expression in the zebrafish gut for the HBGA Lewis X [Le$^X$, $\alpha$(1,3)-fucose] and core fucose [$\alpha$(1,6)-fucose]. Costaining of HuNoV-infected larvae proved colocalization of Le$^X$ and to a lower extent core fucose with the viral capsid protein VP1, indicating the presence of fucose residues on infected cells. Upon blocking of fucose expression by a fluorinated fucose analogue, HuNoV replication was strongly reduced. Furthermore, by comparing HuNoV replication in conventional and germfree zebrafish larvae, we found that the natural zebrafish microbiome does not have an effect on HuNoV replication, contrary to earlier reports about the human gut microbiome. Interestingly, monoassociation with the HBGA-expressing *Enterobacter cloacae* resulted in a minor decrease in HuNoV replication, which was not triggered by a stronger innate immune response. Overall, we show here that fucose has an essential role for HuNoV infection in zebrafish larvae, as in the human host, but their natural gut microbiome does not affect viral replication.

**IMPORTANCE** Despite causing over 700 million infections yearly, many gaps remain in the knowledge of human norovirus (HuNoV) biology due to an historical lack of efficient cultivation systems. Fucose-containing carbohydrate structures, named histo-blood group antigens, are known to be important (co)receptors for viral entry in humans, while the natural gut microbiota is suggested to enhance viral replication. This study shows a conserved mechanism of entry for HuNoV in the novel zebrafish infection model, highlighting the pivotal opportunity this model represents to study entry mechanisms and identify the cellular receptor of HuNoV. Our results shed light on the interaction of HuNoV with the zebrafish microbiota, contributing to the understanding of the interplay between gut microbiota and enteric viruses. The ease of generating germfree animals that can be colonized with human gut bacteria is an additional advantage of using zebrafish larvae in virology. This small animal model constitutes an innovative alternative to high-severity animal models.

**KEYWORDS** glycans, human norovirus, zebrafish, gut microbiota

Address correspondence to Joana Rocha-Pereira, joana.rochapereira@kuleuven.be.

The authors declare no conflict of interest.

Human norovirus (HuNoV) is the major cause of acute gastroenteritis worldwide across all age groups, with an estimated 700 million yearly infections resulting in around 220,000 deaths and an additional economic burden up to $60 billion (1). The

most affected groups are the elderly, young children (<5 years), and immunocompromised patients, who have a higher risk for severe symptoms and complications or even chronic norovirus infections (1, 2). HuNoVs are small (+)-sense single-stranded RNA viruses from the family *Caliciviridae* divided into genetically distinct genogroups (GI-X) and further subdivided into 49 genotypes based on the amino acid sequence of their major capsid protein (VP1) (3). The GII.4 genotype is responsible for the majority of sporadic cases and outbreaks.

The difficulties of cultivating HuNoV in cell culture and a lack of small animal models have hindered research on HuNoV biology in recent decades (4, 5). A significant knowledge gap that remains is the identity of the cellular receptor for HuNoV, as well as a complete understanding of HuNoV entry in the host cells. For murine norovirus (MNV), the protein CD300lf has been identified as the main cellular receptor, yet this is not a functional receptor for HuNoV (6). However, it has been extensively shown that histo-blood group antigens (HBGAs), a family of carbohydrates, function as attachment factors and likely as coreceptors for HuNoV (7, 8). HuNoV attaches to HBGAs via glycan-binding pockets in the P2 subdomain of VP1 (9–11). However, HBGA expression in mammalian cell lines such as Caco-2 is not sufficient for the culture to support robust HuNoV replication (4, 12). In humans, HBGAs are found as terminal residues on glycolipids and glycoproteins expressed on red blood cells or mucosal epithelial cells or secreted in biological fluids (e.g., saliva, milk, and intestinal contents) (13). HBGA synthesis is controlled by several gene families encoding glycosyltransferases, of which the fucosyltransferases (*FUT*) are responsible for the incorporation of fucoses. There are two major families of HBGA antigens: secretor antigens and Lewis antigens, respectively, controlled by *FUT2*, an $\alpha$(1,2)-fucosyltransferase, and *FUT3*, an $\alpha$(1,3/4)-fucosyltransferase (14). Around 20% of the worldwide population have nonsense or missense mutations in their *FUT2* gene, reducing the secretion of HBGAs (individuals are thus called "nonsecretors"). Interestingly, challenge and outbreak studies have shown that nonsecretor individuals have a strongly reduced susceptibility to GII.4 infection (15). Moreover, expression of secretor antigens was shown to be required for successful GII.4 replication in the recently established organoid model for HuNoV. Stem-cell-derived human intestinal enteroids originating from nonsecretor individuals were not permissive for GII.4 infection and showed a reduced replication for other genotypes such as GII.3 (16, 17).

Recently, we have established a robust small animal model for HuNoV in zebrafish larvae (18, 19). Multiple genotypes, including the pandemic GII.4 genotype, are able to infect and replicate in zebrafish larvae after injection of virus in the yolk, which is their food reservoir, thus mimicking the natural route of infection. Zebrafish (*Danio rerio*) are widely used vertebrate model organisms and very attractive to explore in virology given their optical transparency, small size, and genetic similarity to humans (82% of human disease-associated genes have a zebrafish orthologue) (20). In addition to HuNoV, zebrafish have been shown to be permissive to herpes simplex, chikungunya, and human influenza A virus, demonstrating susceptibility to viruses with a diverse range of cellular receptors (21). Moreover, zebrafish have been extensively used as a model to study microbe-host relationships because of the relatively easy and standardized method to rear germfree larvae (22–24). Although there is a significant difference in bacterial composition of the gut microbiota between zebrafish and humans, germfree larvae can be successfully colonized with human gut bacteria through monoassociation or human fecal transplantation (25, 26).

Interestingly, HuNoV has a complex relationship with the host microbiome, with both pro- and antiviral effects reported (27). Certain gut bacterial species also express HBGA-like structures to which HuNoV can attach (28). In B-cell *in vitro* cultures, the addition of *Enterobacter cloacae*, an example of such HBGA-expressing bacteria, was shown to enhance HuNoV replication (29). Moreover, studies in mice showed that removal of the gut microbiome via antibiotic treatment prevented acute and persistent MNV infection through disruption of the interferon-$\lambda$ (IFN-$\lambda$) homeostasis (30).

Alternatively, an increase of *Lactobacillus* spp. in the mouse microbiome through retinoic acid administration reduced MNV replication (31).

Previously, we reported on the susceptibility of zebrafish to infection with various GI and GII genotypes and detected viral antigens in the zebrafish intestines, as well as in the caudal hematopoietic tissue (19). This largely overlaps with what was reported in humans, in which viral antigens were detected in enterocytes of the intestinal epithelium and myeloid cells (32, 33). However, details on why zebrafish are susceptible to HuNoV infection are elusive. We thus investigated whether a conserved mechanism of entry based on fucose attachment is shared between zebrafish and the human host.

Zebrafish have five fucosyltransferase orthologues to human *FUT* genes: *fut7* to *fut11*, encoding either $\alpha(1,3)$- or $\alpha(1,6)$-fucosyltransferases, yet $\alpha(1,2)$-fucosyltransferases are not known to be expressed in zebrafish (34, 35). The expression of HBGAs in zebrafish is therefore genetically limited to the Lewis antigens and more specifically the Lewis X (Le$^x$) antigen. Le$^x$ expression in zebrafish larvae and adults has been confirmed through mass spectrometry and high-performance liquid chromatography (HPLC) (36, 37). The expression of Fut8, the only $\alpha(1,6)$-fucosyltransferase in zebrafish and humans, in zebrafish intestines also infers the presence of core fucose (36, 37), which, in contrast to the terminally expressed Le$^x$, is found directly on the stalk of glycoproteins and glycolipids. Contrary to HBGAs such as Le$^x$, core fucose is not known to be an attachment factor for, or directly interact with, human viruses. However, transfection of the hepatitis B virus (HBV) genome in cells resulted in an increase in core fucosylation, which in turn increased the endocytosis of HBV pseudovirus, potentially by changing the glycosylation of the HBV receptor (38).

For both Le$^x$ and core fucose, studies showing the specific localization and abundance of fucose residues at the surface of the larval intestine at the time point when this organ is completely formed and becomes functional are lacking. Hence, we here investigated the expression of terminal and core fucose in the zebrafish intestine and whether this is required for the start of a HuNoV infection. We next investigated whether the zebrafish microbiome has an enhancing effect on HuNoV replication by comparing replication yields in germfree zebrafish with those with a natural microbiota.

## RESULTS

**HuNoV antigens are present in intestinal cells expressing fucose residues at their surface.** To visualize the expression of Le$^x$ and core fucose in the intestinal tract of zebrafish larvae, we performed whole-mount immunohistochemistry on larvae at 5 days post fertilization (dpf), when the intestinal tract is fully developed and functional. While a Le$^x$-specific antibody was used to stain for Le$^x$, core fucose was stained with a fluorescein-tagged *Aleuria aurantia* lectin [AAL, specific for the $\alpha(1,6)$-core fucose]. Core fucose was found uniformly spread in the intestinal bulb and in strong individual spots along the posterior intestinal tract (Fig. 1A to C). Le$^x$ was found in clearly defined spots along the intestinal tract and in the cells of the ventral and tail fin, as well as around the eye (Fig. 1D and E). Next, to determine whether the HuNoV infection of the intestinal epithelium is closely related to the presence of fucose at the cell surface, we assessed whether HuNoV antigens (specifically the surface capsid protein VP1) colocalized with such fucose residues. To that end, whole-mount stainings were performed on 5 dpf larvae, 1 day after infection with HuNoV GII.4 for either Le$^x$ or core fucose (green) together with a HuNoV VP1-specific antibody (red). Viral antigens were detected in the epithelial cell lining of the midintestines and in the intestinal bulb (Fig. 1F and G). Colocalization was determined based on the Manders' coefficient (Fig. 1H) (39). Although we could detect colocalization for both fucose residues, Le$^x$ has a higher colocalization coefficient than core fucose. Additionally, in the HBGAs binding assays we performed, HuNoV GII.4 virus-like particles (VLPs) bound best to Le$^x$, while positive recognition was also found for blood groups A and B, Le$^b$, type 1 H antigen, Le$^y$, and sialyl-Le$^x$ (sLe$^x$) (Fig. S1). Together, these results show that Le$^x$ and core fucose are abundantly expressed in the intestinal tract of 5 dpf zebrafish larvae and

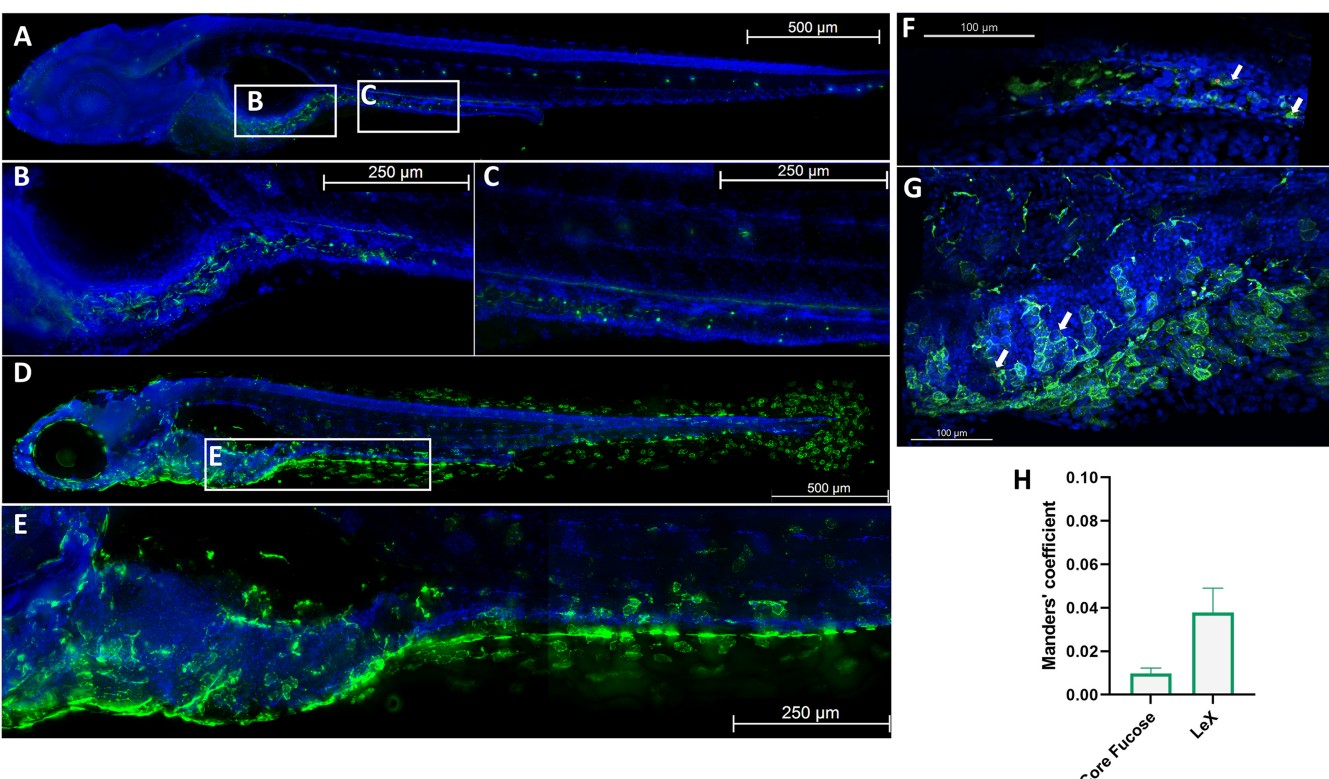

**FIG 1** Lewis X (Le^x) and core fucose are expressed in 5-days post fertilization (dpf) zebrafish larvae. (A to C) Whole-mount immunohistochemistry staining in 5 dpf larvae of core fucose stained with fluorescein isothiocyanate (FITC)-labeled *A. aurantia* lectin (AAL, green) and counterstained with Hoechst 33342 (blue) at ×10 (A) and ×20 (B) magnification in the intestinal bulb and ×20 magnification along the posterior intestine (C). (D, E) Whole-mount immunohistochemistry staining in 5 dpf larvae of Le^x stained with anti-LeX primary antibody (green) and Hoechst 33342 (blue) at ×10 (D) and ×20 (E) magnification, representative image of Le^x staining along the intestinal tract. (F, G) Whole-mount immunohistochemistry staining of human norovirus (HuNoV)-infected zebrafish larvae at ×20 magnification costained with an anti-VP1 antibody (red) and FITC-AAL (F) or Le^x (G), with Hoechst 33342 as counterstaining. White arrows, colocalizing viral particles. (H) Tresholded Manders' colocalization coefficient of HuNoV VP1 with fucose residues in the gastrointestinal tract of 5 dpf larvae (*n* = 5). Tresholded Manders' coefficients are calculated with Imaris colocalization software. For panels A to E, adjacent tile pictures were merged together using the mosaic merge function of the LAS X software.

that HuNoV, as in humans and other models, attaches and enters cells expressing HBGAs, in this case Le^x, on their surface.

**Expression of fucose is required for efficient HuNoV replication in zebrafish.** Next, we verified whether fucose residues are required for HuNoV infection of the zebrafish intestine, in addition to localizing on the surface of infected intestinal cells. To that end, we removed the Le^x fucose and core fucose residues from the glycoproteins on the zebrafish host cells and assessed the effect of this removal on viral replication. First, we attempted to cleave of Le^x fucose and core fucose by injecting two bacteria-encoded fucosidases: AfcB, an α(1,3)-fucosidase, and AlfC, an α(1,6)-fucosidase, cleaving off Le^x fucose and core fucose, respectively (40, 41). AfcB and AlfC were either injected directly into the lumen of the developing intestinal tract of 3 dpf larvae, or in the pericardial sack, which is connected to the blood circulation. Efficacy of the enzymatic cleavage was assessed concurrently by both quantifying viral replication and whole-mount staining. The respective larvae were then infected with HuNoV GII.4 (~10^4 viral RNA copies) on the next day (4 dpf). Viral replication was quantified by quantitative reverse transcription-PCR (RT-qPCR) using pools of 10 larvae at 1 day post infection (dpi). Importantly, the AlfC enzyme was coinjected with endoglycosidase H (endoH), which cleaves the diacetyl-chitobiose core of glycoproteins, making the core fucose of the glycoproteins more exposed and thus more accessible for cleavage by AlfC. However, quantification of the signal showed no difference in fucose expression in enzyme-treated and nontreated larvae, suggesting that no sufficient active enzyme reached the intestinal tract to efficiently cleave fucose residues (Fig. S2A and B). Hence, no reduction in viral replication was

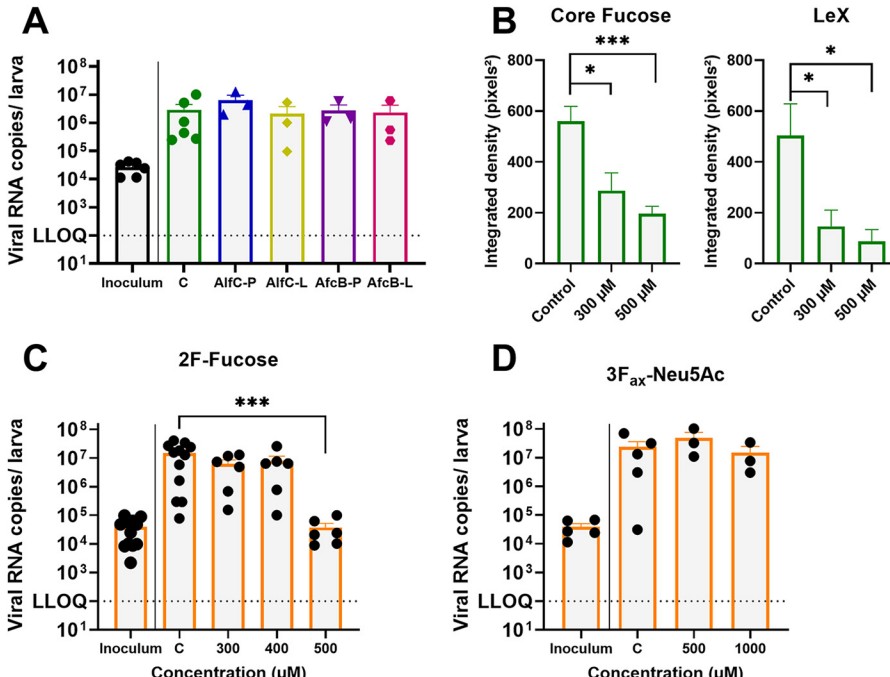

**FIG 2** Inhibition of fucosyltransferases reduces HuNoV replication. (A) Viral RNA copies of HuNoV at 1 day postinfection (dpi). The larvae were injected with AlfC or AfcB fucosidase at 3 dpf and infected with GII.4 HuNoV at 4 dpf ($n$ = 3-5). P, pericardial injection; L, lumen injection. (B) Fucose expression in zebrafish larvae treated with 500 $\mu$M 2F-fucose. Integrated density was quantified in a region of interest comprising the entire gastrointestinal tract with ImageJ. (C) Viral RNA copies of HuNoV per zebrafish larva at 1 dpi. The larvae were immersed in 2F-fucose from 0 to 4 dpf and infected with a GII.4 stool sample at 4 dpf ($n$ = 6 to 12). (D) Viral RNA copies of HuNoV per zebrafish larva at 1 dpi. The larvae were immersed with 3F$_{ax}$-peracetyl-Neu5Ac from 0 to 4 dpf and infected with a GII.4 stool sample at 4 dpf ($n$ = 3 to 5). In panels A, B, and D, the bars represent viral RNA levels/zebrafish larvae, quantified by quantitative reverse transcription-PCR (RT-qPCR). The mean values $\pm$ standard error of the mean (SEM) are presented. Mann-Whitney tests were used. ***, $P \leq 0.001$; **, $P \leq 0.01$; *, $P \leq 0.05$. The dotted line represents the lower limit of quantification (LLOQ).

observed after treatment with either enzyme (Fig. 2A). Prior to injection, we confirmed enzymatic activity *in vitro* as previously described (40, 41).

Alternatively, we exposed the larvae to a range of concentrations of a fluorinated peracetylated fucose analogue, 2F-fucose (2F-F), which inhibits the activity of FUT4 [$\alpha$(1,3)-fucosyltransferase], FUT7 [$\alpha$(1,3)-fucosyltransferase], and FUT8 [$\alpha$(1,6)-fucosyltransferase] (42), hence resulting in a strongly reduced addition of fucose to the host cell surface. The 2F-F was added to the swimming water of dechorionated embryos from 0 dpf onwards and refreshed daily. At 4 dpf, the larvae were infected with HuNoV GII.4 and further kept in fresh water without compound. To confirm that treatment resulted in successful inhibition of fucose expression in the gastrointestinal tract, immunohistochemistry was performed. The signal of Le$^x$ and core fucose expression was reduced by 83 and 65%, respectively, in the intestinal tract of larvae treated with 500 $\mu$M 2F-F, the highest tested concentration (Fig. 2B) (Fig. S3A to D). Although treatment with 2F-F resulted in developmental anomalies such as increased body curvature, absence of swim bladder inflation, and reduced yolk uptake, this effect was not concentration-dependent, and the larvae had no other signs of disease (normal cardiac rhythm, blood circulation, etc.) (Fig. S4A to C). Still, this phenotype highlights that fucose is important for larval development, as described in mammals (43). Next, we infected 2F-F-treated larvae with a GII.4 HuNoV at 4 dpf and quantified viral replication at 1 dpi (5 dpf). While treatment with 300 or 400 $\mu$M 2F-F did not seem to affect viral replication, treatment with 500 $\mu$M 2F-F resulted in a strong 2.6 log$_{10}$ reduction in viral replication compared to the control group (Fig. 2C). This shows that the presence of

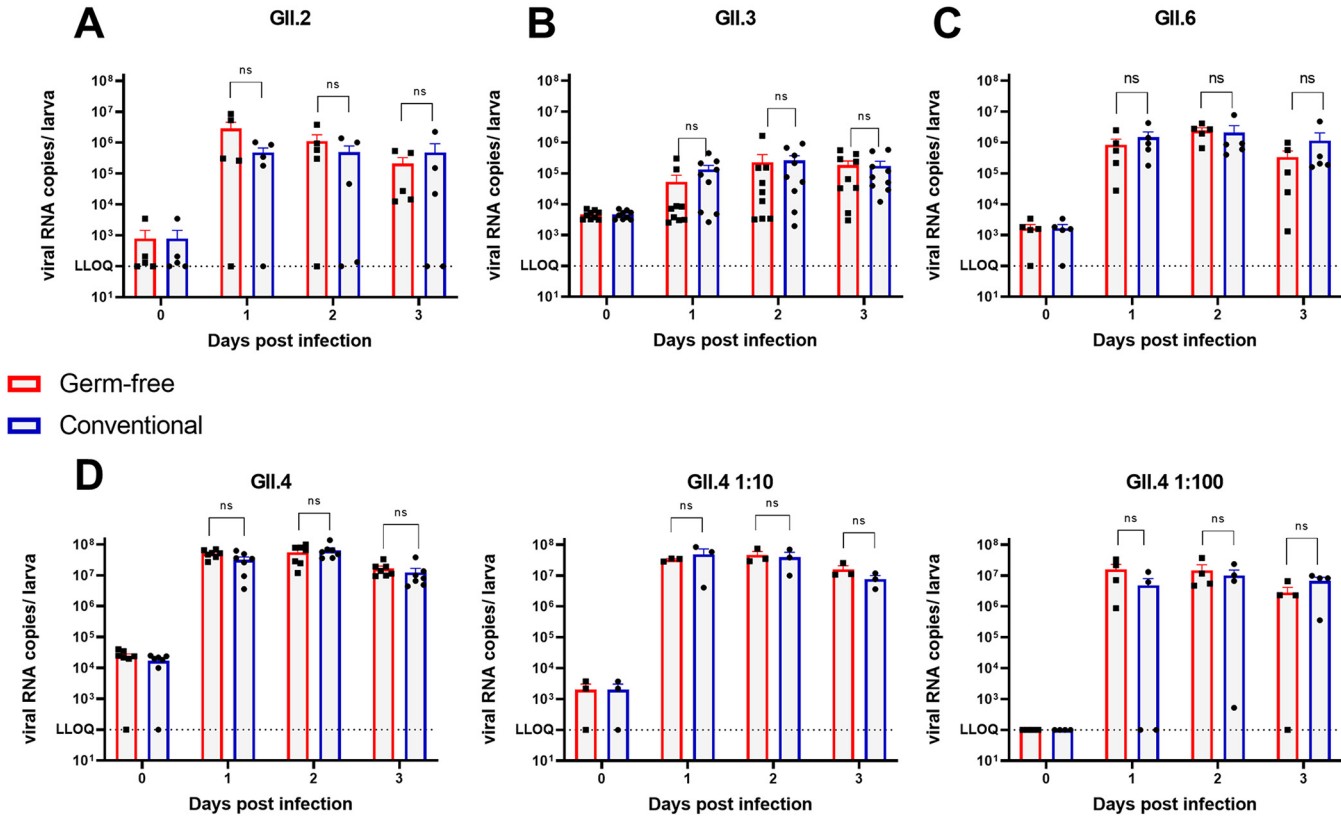

**FIG 3** Germfree status of larvae does not affect HuNoV replication. (A to D) HuNoV replication in germfree and conventional larvae with genotypes GII.2 (A), GII.3 (B), GII.6 (C), and GII.4 (D). Calculated inocula (3 nL per zebrafish larva) are $1.5 \times 10^2$ (A), $8.2 \times 10^3$ (B), $1.3 \times 10^3$ (C), and $1.2 \times 10^4$, $1.2 \times 10^3$, and $1.2 \times 10^2$ (D) viral RNA copies. For all graphs, the bars represent viral RNA levels/zebrafish larva, quantified by RT-qPCR. The mean values $\pm$ SEM are presented. Mann-Whitney tests were performed. The dotted line represents the LLOQ. ns, not significant.

fucose is a necessary requirement for HuNoV to be able to replicate in zebrafish larvae, resembling infection in humans.

Sialic acids are also commonly found in glycan structures and have been suggested to serve as a ligand and attachment factor for HuNoV and MNV, facilitating cell entry, like fucose (44, 45). To verify whether sialic acids functions as an attachment factor for HuNoV in zebrafish larvae, we used the fluorinated analogue $3F_{ax}$-NeuAc (3F-N), which inhibits ST3Gal and ST6Gal I enzymes (42). Contrary to 2F-F, we observed no developmental toxicity after a 4-day treatment with 3F-N, not even when we exposed the larva to higher concentrations of 1,000 $\mu$M (Fig. S4D). Additionally, no difference in viral replication in 3F-N-treated larvae was noted (Fig. 2D).

**Zebrafish gut microbiota does not enhance HuNoV replication.** The human gut microbiota has been shown to have proviral effects via direct and indirect interactions with HuNoV. Addition of the HBGA-expressing *E. cloacae* restored viral replication in B cells, suggesting that attachment to bacterial HBGAs enhances HuNoV infection. Moreover, microbiota has also been shown to counteract the host innate immune response in mice, facilitating MNV infection (30). Hence, we wanted to assess whether the zebrafish gut microbiota could enhance HuNoV replication.

The axenic, *ex utero* development within a protective chorion of zebrafish larvae allows for a relatively straightforward generation of germfree (GF) larvae (23). To generate GF larvae, early-stage (0 hours post fertilization) embryos were consecutively washed in solutions of antibiotics, povidone-iodine, and bleach and then kept in sterile water to preserve the GF status (Fig. S5A). The GF status was verified and monitored over time by growing (an)aerobic bacterial cultures of sampled zebrafish (Fig. S5B) and by quantifying the conserved 16S rRNA gene in sampled zebrafish larvae and swimming water (Fig. S5C). During experiments, the GF status was monitored by sampling zebrafish larvae for

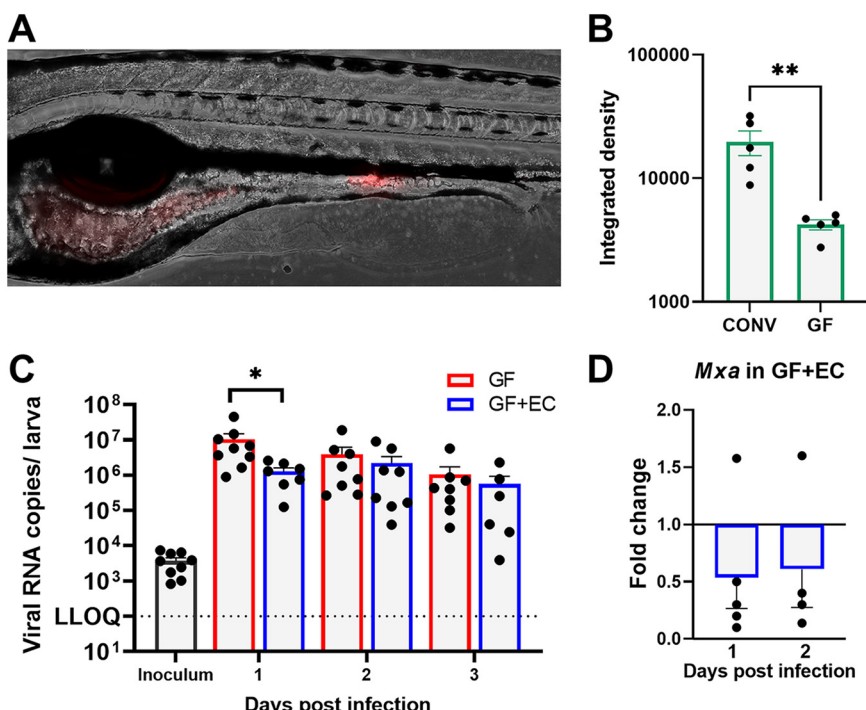

**FIG 4** *E. cloacae* colonization has limited effect on HuNoV replication. (A) Representative image of *E. cloacae* colonization in 4 dpf larvae. Red fluorescent protein (RFP) signal produced by *E. cloacae* is overlaid with a brightfield image. (B) Total bacterial colonization in conventionally raised larvae (CONV) and germfree (GF) larvae represented by quantification of RFP signal. Total RFP signal in individual larvae was quantified with ImageJ. (C) Viral RNA copies of HuNoV in GF or GF + *E. cloacae*-colonized (EC) larvae. The larvae were colonized at 3 dpf and infected with a GII.4 HuNoV-positive stool sample at 4 dpf. The bars represent viral RNA copies/zebrafish larva, quantified by RT-qPCR. The dotted line represents the LLOQ. (D) The effect of HuNoV GII.4 replication on the expression of *mxa* in GF and GF+EC larvae, determined by RT-qPCR. The bars represent the relative expression of *mxa* in HuNoV-infected GF+EC larvae versus infected GF larvae compared to 0 dpi and normalized to the housekeeping genes (*n* = 5). The mean values ± SEM are presented. Mann-Whitney tests were performed to detect significant differences. *, $P \leq 0.05$; **, $P \leq 0.01$.

16S rRNA quantification at the beginning (3 dpf) and end (6 dpf) of the experiment (Fig. S5D). Next, 3 dpf GF and conventionally raised (CONV) larvae were injected with HuNoV of different genotypes. Before injection, the stool samples were filtered through a 0.22-$\mu$m filter membrane to prevent contamination of the zebrafish larvae with residual bacteria. Groups of 10 larvae were harvested every day from 0 to 3 dpi to determine viral loads via RT-qPCR. Contrary to our expectations, we observed no differences in viral replication between GF and CONV larvae with GII.2 (Fig. 3A), GII.3 (Fig. 3B), GII.6 (Fig. 3C), or GII.4 (Fig. 3D) samples. Similar viral RNA levels were reached in both conditions, with a peak of replication at 2 dpi. To make sure that the use of a high inoculum did not mask any subtle differences, 1/10 and 1/100 dilutions of the HuNoV GII.4 sample were injected as well (Fig. 3D). Again, similar viral RNA levels were reached at 1 and 2 dpi in GF and CONV larvae.

To test whether an enhancing effect on the viral replication could be induced by addition of bacteria present in the human gut microbiota, we next colonized GF larvae with red fluorescent protein (RFP)-expressing *E. cloacae* (kind gift from Chang-Hyun Kim, University of Illinois); this bacterium was previously shown to enhance HuNoV replication in B cells. The larvae at 3 dpf were immersed in a solution containing 4 × 10⁸ bacterial cells/mL for 24 h; at this point, the mouth of the larvae is open and thus allows colonization through oral uptake. At 4 dpf, the larvae were washed extensively to remove any *E. cloacae* sticking to the larval skin and in the swimming water. Since the bacteria are RFP labeled, we could confirm successful colonization through live imaging (Fig. 4A). Remarkably, when looking at the RFP signal as a marker for colonization

efficiency, we noticed a reduced signal in colonized GF larvae, compared to colonized CONV larvae. This suggests a reduced efficiency of *E. cloacae* to colonize GF larvae (Fig. 4B). Next, colonized (GF+EC) and noncolonized (GF) larvae were infected at 4 dpf with HuNoV GII.4, and groups of 10 larvae were harvested daily until 3 dpi to determine viral load via RT-qPCR. Surprisingly, we detected a reduced viral titer at 1 dpi in the colonized group (Fig. 4C). This difference was no longer present at 2 and 3 dpi. Hence, as before, we did not detect a HuNoV replication-enhancing phenotype. Since colonization with *E. cloacae* in gnotobiotic pigs also led to a reduced HuNoV replication due to an exacerbated immune response (46), we checked whether a stronger antiviral immune response was present in colonized larvae at 1 and 2 dpi. To that end, we quantified the expression levels of the interferon-stimulated gene *mxa* upon infection of GF+EC versus GF larvae, since we previously showed *mxa* to be upregulated after HuNoV infection (Fig. 4D). The values are represented as the fold change in *mxa* mRNA levels in infected GF+EC larvae versus infected GF larvae, compared to 0 dpi and normalized to housekeeping genes. Contrary to what we expected, we did not observe an increased expression of *mxa* in the GF+EC group after infection. In fact, GF+EC larvae mount a less strong innate immune response upon infection than noncolonized infected GF larvae.

## DISCUSSION

The limited availability of *in vivo* models for HuNoV has left significant knowledge gaps in HuNoV biology regarding cellular receptors and their relationship with gut microbiota in complex models. At the same time, interest in zebrafish larvae as model for human viral infections has increased in recent years. Their unique characteristics such as optical transparency, *ex utero* development, and genetic similarity are good reasons to explain the growing interest in zebrafish (47). We here report the presence of fucose-containing Le$^X$ antigens in the intestinal tract of zebrafish larvae and their essential role during HuNoV infection as a necessary attachment factor for cell entry.

We here visualized the expression pattern in the larval gut for the first time. We observed a strong, clearly outlined signal along the whole intestinal tract for Le$^X$ and in other tissues such as the ventral, dorsal, and tail fins and around the eye, which is in line with reported Le$^X$ expression in the eye and skin of zebrafish larvae (48). Regarding core fucose, we observed strong individual spots along the posterior intestinal tract and a broader signal in the intestinal bulb. Both fucose residues proved to colocalize with the viral capsid protein VP1, indicating that infection of intestinal epithelial cells is closely related to the presence of fucose at the cell surface. As shown by our enzyme-linked immunosorbent assay (ELISA) and previous literature (49, 50), HuNoV virions are able to bind Le$^X$; thus, collectively our data show that HuNoV does enter and replicate in the HBGA-expressing cells of the intestinal tract of zebrafish, as it does in humans.

The absence of a *FUT2* orthologue, or any other $\alpha$(1,2)-fucosyltransferase in zebrafish larvae limits their expression of HBGAs to only two members of the Lewis antigens: Le$^A$ and Le$^X$, for which the difference lies in the glycosidic bond between the fucose and *N*-acetylglucosamine structure, respectively, $\alpha$(1-4) and $\alpha$(1-3) (51). This limited expression pattern is comparable to the nonsecretor phenotype described in humans, in which mutations inactivate the *FUT2* gene. Interestingly, we observe an efficient replication of HuNoV GII.4. This is in contrast with other HuNoV infection models such as the human intestinal enteroid system, in which the absence of a functional *FUT2* gene prevents successful replication of GII.4 HuNoV (17). In addition to HBGAs, norovirus has also been shown to bind other ligands such as heparan sulfate or citrate (52, 53). Hence, it may be that zebrafish larvae express certain additional ligands to which HuNoV is able to bind (54, 55).

Inhibition of $\alpha$(1,3)- and $\alpha$(1,6)-fucosyltransferases by a fluorinated fucose analogue reduced the expression of core fucose and Le$^X$ in the cell surface glycans. Consequently, the absence of Le$^X$ prevented HuNoV from attaching to the required fucose residue and entering the target cell, resulting in a significant reduction of viral replication. The developmental toxicity that was observed due to treatment with 2F-F is most likely due to the

inhibition of the Fut8 enzyme. Indeed, morpholino-mediated knockdown of the *fut8* gene in zebrafish embryos resulted in a similar, curved phenotype due to disrupted midline patterning (56). Despite observing similar phenotypic effects, we found a reduction in viral replication only at the highest concentration of 2F-F. In cell culture, 2F-F treatment inhibited Le$^x$ expression in a clear dose-dependent way; however, core fucose expression was inhibited abruptly (42). It is thus likely that inhibition of the core fucosylation is already efficient and sufficient to induce phenotypical aberrations at lower concentrations, while the slower, dose-dependent effect on Le$^X$ affects viral replication only from a certain threshold concentration.

In contrast to fucosyltransferases, inhibition of sialyltransferases via the fluorinated analogue 3F$_{ax}$-Neu5Ac did not elicit a phenotype, nor did it affect viral replication. Although previous research suggested sialic acid residues as an attachment factor for HuNoV and MNV, these findings were recently attributed to methodological limitations, and new NMR results suggest that neither MNV nor HuNoV P-domains contain binding sites for sialic acid (57). However, compared to humans, zebrafish have a very diverse sialylation pattern containing terminal Neu5Ac and Neu5Gc substrates on their complex glycan structures, resulting in the abundant expression of sLe$^x$ (58). Humans, as well as several other mammalian species, lack the sialic acid Neu5Gc due to evolutionary loss-of-function mutations in the encoding *CMAH* gene (59). Moreover, zebrafish have a higher number of paralogous genes encoding sialyltransferase enzymes than humans due to evolutionary whole-genome duplications (37). Creutznacher et al. (57) also suggested that, while isolated P-domains do not have sialic acid-binding capabilities, whole intact capsids may contain more complex mechanisms to recognize glycans, demonstrated by the fact that HuNoV VLPs can recognize sLe$^x$ as a neoglycoprotein (60). It could be that the diverse sialylation pattern of zebrafish holds unique carbohydrate structures containing Neu5Ac and Neu5Gc sialic acids that function as attachment factor for HuNoV and are not sufficiently inhibited by the fluorinated analogue used here.

In mice, depletion of the intestinal microbiota via antibiotic treatment reduced MNV replication (29). Additionally, in the *in vitro* B cell culture system, filtration of a HuNoV-positive sample decreased viral genome replication compared to the unfiltered sample (29), suggesting there to be microbial components in the murine and human stool and microbiota enhancing viral replication. Indeed, in the B-cell culture, viral replication could be restored by addition of synthetic H-antigen, an HBGA, of which similar structures are expressed on the outer surface of commensal gut bacteria. Taking advantage of the axenic development of zebrafish embryos, which facilitates the generation of germfree animals, we assessed whether the natural microbiota of zebrafish larvae also enhances HuNoV replication. The generation of germfree zebrafish larvae is a well established and commonly used method that we could implement and maintain successfully throughout the experiments. In contrast to our hypothesis that a reduced replication of HuNoV would occur in germfree larvae, we detected no difference in viral replication between germfree and conventional larvae. The expression of HBGA-like structures in bacteria is mostly described in members of the family of Gram-negative *Enterobacteriaceae* (class γ-proteobacteria). In humans, this family is predominantly represented by *Escherichia coli* strains (61). HuNoV VLPs have been shown to bind the secretor-like and Lewis-like HBGA molecules expressed by *E. coli*, resulting in increased resistance to heat treatment of the HuNoV VLPs, implying a protective effect (62). Although the zebrafish microbiota also consists mainly of (γ-)proteobacteria species, including members of the *Enterobacteriaceae* family, the core genera differ from humans. The dominant genera in zebrafish are *Aeromonas* spp., *Pseudomonas* spp., *Shewanella* spp., and *Plesiomonas* spp. (63, 64). Interestingly, *Pseudomonas* species found on lettuce also express HBGA-like structures and were shown to bind different HuNoV genotypes (65). However, this does not necessarily mean an enhancing effect for replication. Additionally, the actual abundance of *Pseudomonas* in our zebrafish

larvae can be variable, as they account for only 0 to 2% of bacterial clones in domesticated zebrafish compared to 18% in recently caught zebrafish (63).

It is possible that the differences in abundance and diversity of bacteria present in zebrafish at the age we use them (3 to 6 dpf) does not add up to the same enhancing effect as in adult mouse models and humans. Indeed, in zebrafish (as in humans), the complete maturation of the gastrointestinal tract requires the presence of commensal microbiota (66, 67). Both reduced proliferation and maturation of enterocytes, and reduced expression of glycoconjugates has been observed in GF zebrafish larvae (68–70). Hence, in our GF zebrafish larva model (like in other GF models), the absence of the microbiota results in an imperfect recapitulation of the gastrointestinal tract of CONV larvae. Moreover, the taxa richness and phylogenetic diversity of the zebrafish commensal gut microbiota changes drastically at 10 dpf (juvenile stage) and again at 75 dpf (adult stage) (64). Additionally, the adaptive immune system, which becomes active only at 4 to 6 weeks postfertilization, may also play a role in maintaining the balance between the host immunity and the natural microbiota (71, 72). The complete maturation of the gastrointestinal tract together with the changes in microbial composition and activation of the adaptive immune response may affect HuNoV replication to a variable degree; hence, it would be of interest to confirm our findings in a juvenile or adult zebrafish model for HuNoV.

Our results suggest that the natural zebrafish microbiota does not have a proviral effect on HuNoV replication at the larval stage. In addition to synthetic H-antigen, the addition of *E. cloacae* expressing HBGA-like structures on its cell surface dose-dependently restored HuNoV replication in B cells. We therefore expected to recapitulate this enhancing effect by colonizing germfree larvae with *E. cloacae*. The localized colonization pattern we observed is in line with a previous report (73) indicating that the colonization was successful. Although mono-associated GF larvae had a lower viral titer at 1 dpi compared to noncolonized GF larvae, we did not observe an overall enhancing effect on HuNoV replication. This contrasts with the results previously observed in B cells. However, in gnotobiotic pigs, similar results were observed upon monoassociation with *E. cloacae*, i.e., reduced viral shedding and lower viral titer in intestinal tissues were reported (46). Viral titer in monoassociated pigs was around $10\times$ lower in the ileum and slightly reduced in the duodenum 3 days postinoculation with HuNoV. While the zebrafish intestine is divided in posterior, mid-, and anterior intestine, its regional functions are comparable to those in mammals. Nevertheless, the small size of the animal would make dissecting the intestine in regions for viral RNA quantification technically challenging. Still, a similar reduction ($10\times$) in viral RNA at 1 dpi was observed in the total larva. Monoassociated pigs also had enlarged Peyer's patches and wider gut-associated lymphoid tissue, indicating enhanced gut immunity. Therefore, the authors suggested that the inhibition of viral replication in the monoassociated pigs might be in part due to an enhanced immune reaction as a result of the *E. cloacae* colonization (46). To investigate whether the *E. cloacae* colonization in zebrafish larvae affected the innate immune response in a similar way, we studied the upregulation of *mxa*, an interferon-stimulated gene known to be upregulated during HuNoV infection in zebrafish (19). However, *mxa* expression in colonized germfree larvae was not increased compared to noncolonized germfree larvae. While *mxa* is an important component of the response to a viral infection, it may be that the bacterial colonization skews the immune response by other affected components and signaling pathways yet still indirectly affects HuNoV replication. Examples of these gut immune responses seen in other bacterial colonizations are increased neutrophil recruitment and increased proinflammatory mRNAs (74–76).

In summary, we show that successful HuNoV infection in zebrafish larvae requires the expression of terminal fucoses as part of HBGAs. Akin to infections in humans and other recently established *in vitro* models for HuNoV, we show a requirement for the presence of fucose residues on the intestinal cell, inferring a conserved mechanism of entry for HuNoV infection of zebrafish. Furthermore, we demonstrate that the zebrafish

microbiota or the presence of HBGA-expression bacteria in the zebrafish intestine does not enhance HuNoV replication in the early larval stage of their development.

## MATERIALS AND METHODS

**Ethics statement.** All zebrafish experiments were approved and performed according to the rules and regulations of the Ethical Committee of KU Leuven (P142/2021) in compliance with the regulations of the European Union (EU) concerning the welfare of laboratory animals as declared in Directive 2010/63/EU. Stool samples positive for HuNoV were obtained from the University Hospital of Leuven (UZ Leuven, Belgium) according to the rules and regulations of the Ethical Committee of KU Leuven (G-2021-4376) and the UZ Leuven (s63536).

**Zebrafish husbandry and maintenance.** Adult wild-type AB zebrafish are housed in the Aquatic Facility at the KU Leuven at a temperature of 28°C with a 14-h light/10-h dark cycle. Fertilized eggs were obtained from adult zebrafish in mating cages, and collected fertilized eggs were maintained in petri dishes (140 × 20.6 mm) with Danieau's solution [1.5 mM HEPES buffer, 0.12 mM MgSO$_4$, 0.18 mM Ca(NO$_3$)$_2$, 0.21 mM KCl, 17.4 mM NaCl, and 0.6 $\mu$M methylene blue] in an incubator, set at 28°C with a 14-h light/10-h dark cycle.

**Generation of germfree zebrafish larvae.** The generation of GF zebrafish was performed as previously reported with slight modifications (24). Briefly, fertilized embryos were kept in antibiotic embryo medium (ABEM) for 6 h at 28°C. For the generation of ABEM, antibiotics were added to sterile, 0.2-$\mu$m membrane-filtered Danieau's solution to a final concentration of 100 $\mu$g/mL ampicillin (Thermo Fisher Scientific, Waltham, MA), 5 $\mu$g/mL kanamycin (Thermo Fisher Scientific, Waltham, MA), and 250 ng/mL amphotericin B (VWR, Radnor, PA). Afterwards, the embryos were gently washed thrice with ~25 mL sterile Danieau's solution in a 50-mL conical tube. Next, the embryos were immersed in 0.05% (wt/vol) povidone-iodine solution (Toronto Research Chemicals, Toronto, Canada) for 75 s followed by another washing step as described before. Finally, the embryos were immersed in 0.003% (vol/vol) bleach solution (Diversey Holdings, Fort Mill, SC) for 20 min. Finally, embryos were washed thrice with ABEM and transferred to a T-75 flask containing sterile Danieau's solution and maintained at 28°C with a 14-h light/10-h dark cycle until infection.

**Validation germfree status.** To validate the sterility of the GF zebrafish larvae before infection, GF larvae swimming water was collected at 2 dpf, and 1 $\mu$L was inoculated on tryptic soy agar (TSA) plates under aerobic and anaerobic conditions at 37°C. To monitor the sterility of the GF zebrafish larvae throughout the experiment, 10 zebrafish larvae and 100 $\mu$L water from each condition (GF and CONV) were harvested separately and in duplicate at 3 dpf (before injection) and 3 dpi. Zebrafish larvae and water samples were cultured on TSA plates as previously mentioned, whereby the zebrafish samples were first homogenized with disposable pellet pestles (VWR, Leuven, Belgium) and then inoculated onto TSA plates. Agar plates were incubated under aerobic and anaerobic up to 96 h.

For quantification of total bacterial DNA, a one-step qPCR against the 16s rRNA gene was performed using the iTaq universal probes one-step kit (Bio-Rad, Hercules, CA); primers and probes used are listed in Table 1. The cycle conditions were as follows: initial denaturation at 95°C for 3 min followed by 40 cycles of amplification (95°C for 15 s, 60°C for 30 s) (Roche LightCycler 96, Roche Diagnostics, Risch-Rotkreuz, Switzerland). For absolute quantification, standard curves were generated using 10-fold dilutions of template DNA of known concentration.

**Bacterial DNA extraction from GF water and larvae.** Bacterial DNA from the harvested zebrafish larvae and water samples was extracted with the QIAamp DNA minikit (Qiagen, Hilden, Germany) according to the manufacturer's protocol with the following adaptations: After addition of ATL lysis buffer and proteinase K, the samples were vortexed for 15 s. Additionally, the duration of the first incubation step at 56°C was set at 3 min, and final elution was done with 100 $\mu$L elution buffer. Extracted DNA was stored at −80°C for long-term storage and −20°C for short-term storage.

**Processing of HuNoV-positive feces samples.** Human feces samples containing HuNoV were acquired anonymously from the existing collection of samples of the University Hospital of Leuven (Belgium). From each stool sample, 100 mg was aliquoted and resuspended in 1 mL of sterile Gibco Dulbecco's phosphate-buffered saline (DPBS; Thermo Fisher Scientific, Waltham, MA) followed by thorough mixing using a vortex mixer and subsequent centrifugation (5 min at 10,000 × $g$). Next, the supernatant was harvested, whereby ~500 $\mu$L was filtered through a sterile 0.22-$\mu$m membrane filter (Merck Millipore, Burlington, MA) using a 1-mL sterile syringe. The supernatants were stored at −80°C.

The virus-DPBS suspensions were used for DNA and RNA extractions, quantification by RT-qPCR, and zebrafish larvae injections. Additionally, 10 $\mu$L of the unfiltered and filtered HuNoV suspensions were aerobically cultured on tryptic soy agar (TSA) plates at 37°C to examine the presence of bacteria. HuNoV RNA was extracted from 50 $\mu$L virus suspension by adding 350 $\mu$L of TRIzol reagent (Thermo Fisher Scientific, Waltham, MA) to the sample, vortexing, and using the Direct-zol RNA miniprep (Zymo Research, Irvine, CA) according to the manufacturer's protocol.

**Compound treatment.** Zebrafish embryos (0 dpf) were treated with 1 mg/mL pronase (*Streptomyces griseus*; Roche Diagnostics, Mannheim, Germany) to remove the chorion. Dechorionated embryos were kept in 1 mL of Danieau's solution, to which 100 mM 2F-peracetyl-fucose (Sanbio, Uden, The Netherlands) or 3F$_{ax}$-Neu5Ac (R&D Systems, Minneapolis, MN) was added to its final concentration. Vehicle control embryos were treated with an equal volume of dimethyl sulfoxide (DMSO). At 1, 2, and 3 dpf compound was refreshed at 4 dpf, and the larvae were washed thoroughly before infection with HuNoV.

**HuNoV injection of zebrafish larvae.** Microinjections were performed as described previously (18). Briefly, 3 or 4 dpf zebrafish larvae were anesthetized by immersion in 0.4 mg/mL tricaine in Danieau's

**TABLE 1** Primers and probes used in quantitative reverse transcription-PCR

| Primer/probe | Forward/reverse | Sequence (5′ to 3′) | Reference |
|---|---|---|---|
| Primer | | | |
| HuNoV | Forward (QNIF2) | ATGTTCAGRTGGATGAGRTTCTCWGA | 80 |
| | Reverse (COG2R) | TCGACGCCATCTTCATTCACA | |
| 16S | Forward | GTGSTGCAYGGYTGTCGTCA | 81 |
| | Reverse | ACGTCRTCCMCACCTTCCTC | |
| mxa | Forward | ATAGGAGACCAAAGCTCGGGAAAG | 82 |
| | Reverse | ATTCTCCCATGCCACCTATCTTGG | |
| 18S | Forward | CGGAGGTTCGAAGACGATCA | 83 |
| | Reverse | TCGCTAGTTGGCATCGTTTATG | |
| $\beta$-Actin | Forward | ATGGATGAGGAAATCGCTG | 82 |
| | Reverse | ATGCCAACCATCACTCCCTG | |
| ef1a | Forward | GCTGATCGTTGGAGTCAACA | 84 |
| | Reverse | ACAGACTTGACCTCAGTGGT | |
| | | | |
| Probe | | | |
| RING2 | | FAM-TGGGAGGGCGATCGCAATCT-TAMRA | 85 |

solution (Sigma-Aldrich, St. Louis, MO). The zebrafish larvae were then positioned in a 2% (wt/vol) agarose mold and infected with HuNoV via a 3-nL microinjection in the yolk. Infected larvae were recovered in Danieau's solution in a six-well plate with up to 20 larvae/well and kept at 32°C in a 14-light/10-h dark cycle for up to 3 dpi.

The GF larvae were oriented inside the laminar airflow hood by using a microscope with external screen (Leica Microsystems, Wetzlar, Germany; DMS1000). Infected GF larvae were kept in sterile Danieau's solution. Each day, up until 3 dpi, 10 zebrafish larvae were harvested in 2 mL RNase- and DNase-free microtubes prefilled with 1.4-mm ceramic beads (Omni International, Kennesaw, GA; 19-627-1000) and stored at −80°C.

**Bacterial fucosidase synthesis.** The *Lactobacillus casei* fucosidase AlfC with selectivity for the fucose $\alpha$(1,6)-linkage was produced in *E. coli* as a His-tagged protein and purified by affinity chromatography as previously described (41). The AfcB enzyme from *Bifidobacterium bifidum* was also produced in *E. coli* as previously described (40). We are grateful to Takane Katayama for providing the *E. coli* strains expressing the AfcB $\alpha$-L-fucosidase. After production and purification of both fucosidases, they were dialyzed in PBS and stored at 1 mg/mL at −20°C until use.

**Fucosidase injection of zebrafish larvae.** For injection into the intestinal lumen, 3 dpf larvae were anesthetized in 0.4 mg/mL tricaine in Danieau's solution and placed on a petri dish lid (100 mm × 15 mm) in a drop of tricaine solution. As much liquid as possible was removed while orientating the larvae in a way that their right or left side was facing up. With a fine capillary needle, 2 nL of AlfC + endoH (New England Biolabs, Ipswich, MA) or AfcB was injected in the developing lumen. Injection into the pericardial sac was done as described before (18).

**Viral RNA extraction from infected zebrafish larvae.** First, 350 $\mu$L of TRIzol reagent (Thermo Fisher Scientific, Waltham, MA) was added to the microtubes containing the harvested zebrafish larvae. The samples were then homogenized for 10 s at 6,300 rpm (Bertin Technologies, Montigny-le-Bretonneux, France).

After, the homogenates were centrifuged (5 min, 8,000 × *g*), and the supernatant was transferred to a sterile Eppendorf tube. An equal volume of absolute (98 to 100%) ethanol was then added to the homogenate followed by the extraction of the viral RNA using the Direct-zol RNA miniprep kit (Zymo Research, Irvine, CA) according to the manufacturer's protocol. The extracted RNA was stored at −80°C for long-term storage and −20°C for short-term storage. Viral RNA was detected using GII HuNoV specific primers (Table 1).

**VLP production.** GII.4 2012 Sydney VLPs (GenBank accession number MN248513.1) were produced as mixtures of VP1 and VP2 proteins. Coding sequences were ordered as synthetic genes with codon usage optimization for insect cells expression from Gene-ART technologies (Thermo Fisher Scientific, Waltham, MA). Both synthetic genes were cloned in pFastbac 1, and each plasmid was transformed into DH10Bac competent cells. The recombinant baculoviruses were produced following the manufacturer's recommendations (Thermo Fisher Scientific, Waltham, MA). Briefly, Sf9 cells were grown in SF900 SFMII medium supplemented with antibiotics and 1% Pluronic (Sigma-Aldrich, St. Louis, MO) in suspension in spin flasks at 120 rpm and 27°C. The cells were subcultured to a density of 3 × $10^5$ cells/mL and infected at a multiplicity of infection of 2 when they reached a density of 1.5 × $10^6$ cells/mL. The cell supernatants were removed after 7 days, and VLPs were purified from the medium by ultracentrifugation. Briefly the supernatant (500 mL) was clarified by centrifugation at 3,000 × *g* for 30 min at 4°C. Baculovirus in the supernatant was pelleted by ultracentrifugation at 100,000 × *g* for 1 h at 4°C with a R25ST rotor in a CR30NX centrifuge (Himac, Ibaraki, Japan). The supernatant was stirred overnight at 4°C with 15% PEG 8,000 and NaCl 0.3 M and centrifuged at 10,000 × *g* for 30 min. The pellet was gently resuspended with PBS, and the VLPs were ultracentrifuged overnight at 197,000 overnight *g* at 4°C with a SW-41 rotor in a L8-M ultracentrifuge (Beckman, Brea, CA). The pellet was resuspended in PBS, and the VLPs were purified by size-exclusion chromatography on a Superdex S-200 column (GE Healthcare, Chicago, IL) column

with PBS. The VLPs were analyzed by transmission electron microscopy (Fig. S1) as previously described (77) and stored at 4°C in PBS.

**VLPs glycan-binding assays.** A panel of sugar antigens conjugated to human serum albumin (HSA) including Lewis A, Lewis B, Lewis X, sialyl-Lewis X, Lewis Y, H type-1, GMI ganglioside, and blood group A and B trisaccharides were purchased from Isosep Ab (Sweden). These neoglycoproteins contain multiple carbohydrates linked to HSA lysines ideal to multivalent presentation of glycans; 96-well microtiter plates (Corning, Glendale, AZ) were coated with the HSA conjugated oligosaccharides (1 $\mu$g/mL) in 0.1 M carbonate/bicarbonate buffer (pH 9.6) and incubated for 1 h at 37°C. After functionalization, the plates were washed once with PBS containing 0.05% of Tween 20 (PBS-T) and then were blocked with 3% BSA in PBS for another hour at 37°C. The plates were washed once with PBS-T, and the GII.4 2012 Sydney VLPs were added (10 $\mu$g/mL in PBS-T) and incubated at 4°C overnight. After three washes with PBS-T, mouse polyclonal antibody anti-GII.4 2012 Sydney VLPs (1:2500) made in-house were added, and the plates were incubated 1 h at 37°C. Then, the plates were washed three times with PBS-T and incubated for 1 h at 37°C with 1:10,000 dilution of horseradish peroxidase (HRP)-conjugated goat anti-mouse (Thermo Fisher Scientific, Waltham, MA). After three washes with PBS-T, the binding was detected using SigmaFast OPD (Sigma-Aldrich, St. Louis, MO) according to the manufacturer's instructions. The absorbance at 492 nm was registered by a MultiScan microplate reader (Thermo Fisher Scientific, Waltham, MA). All the binding assays were performed in triplicate, and the absorbance of unconjugated HSA used as a negative control was subtracted from each glycan.

**Bacterial colonization.** The RFP-*E. cloacae* (kind gift from Chang-Hyun Kim, University of Illinois) were cultured in LB liquid medium supplemented with 50 $\mu$g/mL kanamycin at 37°C for 24 h and shaking at 250 rpm/min. After 24 h, optical density at 600 nm ($OD_{600}$) of the culture was measured, and the number of cells was calculated and diluted to $4 \times 10^8$ cells/mL. Then, 2 mL of the culture was centrifuged (5,000 $\times$ g for 5 min), and supernatant medium was removed. The pellet was resuspended in 2 mL of Danieau's solution. At 3 dpf, the GF and CONV larvae were exposed to 2 mL of the bacterial solution for 24 h. After incubation, the larvae from each group were washed five times with sterile Danieau's solution to remove the bacteria and subsequently injected with HuNoV as described before.

**Characterization of the innate immune response upon HuNoV infection of germfree larvae.** To generate the cDNA, the ImProm II reverse transcription system (Promega, Madison, WI) was used. Briefly, a total of 1 $\mu$g of extracted RNA was added to 2 $\mu$L of random primers to a total volume of 15 $\mu$L and incubated at 70°C for 5 min, followed by 5 min at 4°C. To this reaction mix was added a total volume of 45 $\mu$L containing 8 $\mu$L of Improm II 5$\times$ reaction buffer, 6 mM $MgCl_2$, 0.5 mM deoxynucleoside triphosphate, and 1 $\mu$L of Improm II reverse transcriptase, followed by an incubation at 25°C for 5 min, 37°C for 1 h, and 72°C for 15 min. A qPCR was performed with 4 $\mu$L template cDNA using the Sso Advanced Universal SYBR green Supermix (Bio-Rad, Hercules, CA), 600 nM forward and reverse primers for *mxa*, and the housekeeping genes $\beta$-actin, 18S, and ef1a (Table 1). The cycling conditions were as follows: polymerase activation at 95°C for 3 min followed by 40 cycles of denaturation at 95°C for 15 s, annealing at 55°C, and extension at 72°C for 30 s. The expression levels of colonized GF larvae were compared to noncolonized GF zebrafish larvae and normalized to the housekeeping genes by determining the fold induction of the expression, according to the $2^{(-\Delta\Delta C(T))}$ method (78). Values below 1 indicate a lower expression of *mxa* in GF+EC compared to GF; values above 1 indicate a higher expression.

**Immunohistochemistry staining.** Whole-mount zebrafish staining was performed based on a protocol by Jean-Pierre Levraud (personal communication). Briefly, 5 dpf anesthetized zebrafish larvae were fixed for 2 h in 4% paraformaldehyde in PBS. After fixation, the larvae were rinsed with autoclaved MilliQ water for 30 min while shaking. Next, the larvae were immersed in cold 100% acetone for 20 min at −20°C. Thereafter, the larvae were washed thrice with 0.01% Tween 20 in Gibco Hanks' balanced salt solution (HBSS) (Thermo Fisher Scientific, Waltham, MA) and permeabilized in 1.5 mg/mL collagenase (Sigma-Aldrich, St. Louis, MO) in HBSS + 0.01% Tween + 5 mM $CaCl_2$ for 2 h at room temperature. Next, the larvae were blocked with 10% sheep serum (Sigma-Aldrich, St. Louis, MO) in PBSDT (PBS + 0.1% Triton + 1% DMSO) for 2 h at room temperature. After, primary antibody in blocking solution was added and kept overnight at 4°C. After, the larvae were washed four times for 30 min in PBSDT and blocked for 1 h in blocking solution, after which secondary antibody in blocking solution was added and kept overnight at 4°C. Next, the larvae were washed twice for 15 min with PBSDT followed by a counterstaining with 2 $\mu$g/mL Hoechst 33352 (Thermo Fisher Scientific, Waltham, MA) in PBT (PBS + 0.1% Tween 20) for 30 min. Finally, the fish were washed twice for 15 min and four times for 30 min in PBT. After staining, the zebrafish larvae were kept in 80% (vol/vol) glycerol in PBT and stored at 4°C until use.

The primary antibodies/lectins and dilutions used were fluorescein-labeled *A. aurantia* lectin (1:250, Vector Laboratories, Burlingame, CA), anti-Le$^X$/SSEA-1 (1:100, MC-480 (SSEA-1) deposited into the DSHB by Solter, D./Knowles, B.B (DSHB Hybridoma Product MC-480 [SSEA-1]), and anti-VP1 (1:100)). The secondary antibodies used at 1:400 dilution were goat anti-mouse IgM $\mu$-chain Alexa Fluor 488 (Abcam, Cambridge, UK) and goat anti-mouse Alexa Fluor 596 (Abcam, Cambridge, UK).

**Microscopy and image analysis.** Representative images for Le$^X$/core fucose costaining with anti-VP1 antibody (Fig. 1) were taken with the Andor Dragonfly 200 series high-speed confocal platform system (Oxford Instruments, Abingdon, UK) connected to a Leica DMi8 (Leica Microsystems, Wetzlar, Germany). Image processing and colocalization were done with the Imaris analysis software (Oxford Instruments, Abingdon, UK).

All other images were taken with a Leica DMi8 inverted fluorescence microscope (Leica Microsystems, Wetzlar, Germany) and processed with the associated Leica Application Suite X (LAS X) software. To visualize the zebrafish, adjacent tile pictures (10% overlap) were merged together into one picture using the mosaic merge function of the LAS X software. The larvae were imaged in z-stacks and subsequently

processed by the 3D-Deconvolution software of LAS X and presented as maximum projections. Quantification of fluorescent signal was done with open-source FIJI-ImageJ (79).

**Statistical analysis.** Statistical analyses were performed with GraphPad Prism 9. The methods used are indicated in the figure legends.

## SUPPLEMENTAL MATERIAL

Supplemental material is available online only.
**SUPPLEMENTAL FILE 1**, PDF file, 0.6 MB.

## ACKNOWLEDGMENTS

We thank Chang-Hyun Kim (University of Illinois) for the RFP-labeled *E. cloacae* and Jean-Pierre Levraud (Institut Pasteur) for sharing his protocol on whole-mount immunohistochemistry. We very much appreciate the help of the KU Leuven aquatic facility for the breeding of and advice on the zebrafish larvae. We thank Pedro Elias Marques and Matheus Silvério De Mattos (Laboratory of Molecular Immunology, KU Leuven) for the use and advice on the confocal microscope and Imaris software. We also very much thank Emma Roux and Jana Van Dycke for their technical assistance and careful reading of the manuscript.

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
