## [Reviewer comments · Microbiology Spectrum]

Microbiology Spectrum

The role of histo-blood group antigens and microbiota in human norovirus replication in zebrafish larvae

Arno Cuvry, Roberto Gozalbo-Rovira, Dufie Strubbe, Johan Neyts, Peter de Witte, Jesús Rodríguez-Díaz, and Joana Rocha-Pereira

Corresponding Author(s): Joana Rocha-Pereira, Rega Institute for Medical Research, KU Leuven

Review Timeline:

Submission Date:	August 12, 2022
Editorial Decision:	September 4, 2022
Revision Received:	September 26, 2022
Accepted:	September 30, 2022

Editor: Robert de Vries

Reviewer(s): The reviewers have opted to remain anonymous.

Transaction Report:

DOI: <https://doi.org/10.1128/spectrum.03157-22>

September 4, 2022

Dr. Joana Rocha-Pereira
Rega Institute for Medical Research, KU Leuven
Minderbroedersstraat 10
Leuven 3000
Belgium

Re: Spectrum03157-22 (The role of histo-blood group antigens and microbiota in human norovirus replication in zebrafish larvae)

Dear Dr. Joana Rocha-Pereira:

Thank you for submitting your manuscript to Microbiology Spectrum. As you will see your paper is very close to acceptance. Please modify the manuscript along the lines I have recommended. As these revisions are quite minor, I expect that you should be able to turn in the revised paper in less than 30 days, if not sooner. If your manuscript was reviewed, you will find the reviewers' comments below.

When submitting the revised version of your paper, please provide (1) point-by-point responses to the issues raised by the reviewers as file type "Response to Reviewers," not in your cover letter, and (2) a PDF file that indicates the changes from the original submission (by highlighting or underlining the changes) as file type "Marked Up Manuscript - For Review Only". Please use this link to submit your revised manuscript. Detailed instructions on submitting your revised paper are below.

Link Not Available

Sincerely,

Robert de Vries

Reviewer comments:

Reviewer #1 (Comments for the Author):

Cuvry et al., have investigated the role of histo-bloodgroup antigens and microbiota in human norovirus replication in zebrafish. The study is of interest because still little is known about the interactions between the microbiome and noroviruses. In addition, the zebrafish is a relatively new model for human norovirus and it finally allows researcher to investigate norovirus infections in a small animal model.

The study is well written, and it is appreciated that also negative data and methods are described.

Minor comments

Please check all abbreviations throughout the manuscript/figure legends/figures, some are never indicated or very late in the manuscript.

Line 72 "reducing expression of secretor antigens" should be "reducing the secretion of HBGA". In non-secretors, HBGA are still expressed on other tissues, and glycans are still secreted, but without the fucose group.

Line 107 I would remove natural

Line 114 ref 38 is not a zebrafish study

Line 116-119 Not sure what the authors mean here. Do you mean that core fucose is important for HBV? Please rephrase

Line 120 In which cases? The studies mentioned in ref 36 and 37?

Line 116 no = not?

Line 202-206 The authors use the word "suggested" twice but these studies (Science 2x) "showed/proved" that this was the case for the cells/animal models that they used, with many controls. So please reword.

Line 241 remove "clear"

Line 270 like it is written now it seems that the ELISA data shows that "HuNoV enters and replicates in HBGA-expressing cells....." But this should be something like "collectively the data presented here shows that HuNoV enters and replicates in HBGA-expressing cells....."

Line 365 The authors compare their findings to those with the B-cell model, however the B-cells do not have glycans that are bound by HuNoV, and without bacteria/synthetic HBGA there is no replication at all. This is very different from the zebrafish model, where there is efficient HuNoV replication and the presence of glycans that are bound by HuNoV.

Reviewer #2 (Comments for the Author):

The manuscript by A. Cuvry et al. successfully demonstrate the expression of alpha1-2 and alpha1-6 fucosylated carbohydrates in zebrafish, as well as they role in HuNoV infection in this animal model. It also illustrates how bacteria expressing HBGA do not enhance HuNoV replication in early larval stage. The experiments are properly designed and executed. The scientific quality of this work is high, and the results are sound for the community. I only have some minor comments:

1- I am very skeptical with the high variability shown in the binding assay in Supplementary Figure 1. Some reasons are:

- I would like to drag the attention of the authors to the crystal structures available for this strain in complex with HBGAs (<https://doi.org/10.1128/JVI.02968-14>) and with milk oligosaccharides (10.1128/JVI.03223-15). A simple superimposition of available crystal structures will show that carbohydrates show very similar (in some cases identical) interactions with the P domains, which strongly suggests similar affinities. As an example, please compare PDBs 4X0C (Lewis X) vs 4OP7 (B-tris).

- In addition, dissociation constants KDs are available for a related strain GII.4 Saga 2006 (GenBank AB447457). Saga and Sydney share a 95% VP1 identity, with 100% conservation of amino acids located in the binding pocket. Comparison of crystal structures of Saga (4X06) and of Sydney (4OP7) in complex with B-tris show again identical interactions. NMR data available for Saga P dimers reveals very similar KDs for A- and B-tris.

- Also, Lea and LeX possess almost identical 3D structural principles, which are in turn responsible for their recognition by hNoV (10.1016/j.coviro.2018.04.007). Why should they exhibit such a large difference in Abs?

In summary, these results need to be taken with care. In my opinion, the differences in Abs observed between HBGAs are likely due to variable multivalent presentation, rather than real affinity. That is, it is highly probably that the short and rigid acetyl-phenylenediamine link used by IsoSep AB to connect the carbohydrates with the HSA is influencing the outcome of the experiment. The authors should at least discuss it in the Supplementary Figure 1.

2- Line 281: What about bile acids? They have been shown to interact in different ways with several hNoV strains. Does it make any sense in the biology of zebrafish larvae?

3- It is always helpful to add the GenBank accession number of the hNoV construct overexpressed, in this case VLPs. To be more precise, I would like to know if the VLPs used in this study and named as "GII.4 2012 Sydney" are in fact Hu/GII.4/Sydney/NSW0514/2012/AU, with GenBank accession number JX459908.1. Please add.

4- Please swap order of appearance in the text of figure 2B and Figure 2C. The order should be the one of appearance in the text. Also correct Fig. 2 if required.

5- Line 219: "harvested each day pi.." Is pi a typo?

Preparing Revision Guidelines

Please return the manuscript within 60 days; if you cannot complete the modification within this time period, please contact me. If you do not wish to modify the manuscript and prefer to submit it to another journal, please notify me of your decision immediately so that the manuscript may be formally withdrawn from consideration by Microbiology Spectrum.

Leuven, September 20th 2022

Dear editor,

Please find below our point-by-point responses to the questions raised by the reviewers:

Reviewer #1:

1. Please check all abbreviations throughout the manuscript/figure legends/figures, some are never indicated or very late in the manuscript.

The text has been thoroughly checked for correct use of abbreviations and corrected where necessary.

2. Line 72: "reducing expression of secretor antigens" should be "reducing the secretion of HBGA". In non-secretors, HBGA are still expressed on other tissues, and glycans are still secreted, but without the fucose group.

We have modified the text according to the reviewer's request.

3. Line 107: I would remove natural

We have modified the text according to the reviewer's request.

4. Line 114: ref 38 is not a zebrafish study

Reference 38 was an error and has been removed.

5. line 116-119: Not sure what the authors mean here. Do you mean that core fucose is important for HBV? Please rephrase

While core fucose is not known to bind any human viruses or function as an important attachment factor like Le^x, core fucosylation has been shown to positively affect HBV cell entry through modulation of HBV-receptor-mediated endocytosis. We have modified the text to clarify the enhancing effect core fucosylation has on HBV infection, according to the reviewer's request.

6. Line 120: In which cases? The studies mentioned in ref 36 and 37?

'In both cases' referred to Le^x and core fucose expression in zebrafish larval intestine. Wording has been adapted for clarification.

7. Line 116: no = not?

We have modified the text according to the reviewer's request.

8. Line 202-206 :The authors use the word "suggested" twice but these studies (Science 2x)

"showed/proved" that this was the case for the cells/animal models that they used, with many controls. So please reword.

We have rephrased the text according to the reviewer's request but kept the term 'suggested' when referring to the mechanism of action as the underlying mechanisms have not been fully elucidated yet.

9. Line 241: remove "clear"

We have modified the text according to the reviewer's request.

10. Line 270: like it is written now it seems that the ELISA data shows that "HuNoV enters and replicates in HBGA-expressing cells....." But this should be something like "collectively the data presented here shows that HuNoV enters and replicates in HBGA-expressing cells....."

Sentence has been modified to clarify our conclusion according to the reviewer's request.

11. Line 365: The authors compare their findings to those with the B-cell model, however the B-cells do not have glycans that are bound by HuNoV, and without bacteria/synthetic HBGA there is no replication at all. This is very different from the zebrafish model, where there is efficient HuNoV replication and the presence of glycans that are bound by HuNoV.

The B-cell model does indeed show little to no replication after removal of bacteria from the sample which is different from our zebrafish model. We do however want to point out in the text that in the B cells, merely addition of (heat-killed) E.cloacae had a potent restorative effect on HuNoV replication. Based on these observations, we expected also an enhancing effect in our zebrafish model, even though there already is a robust replication in conventional and germ-free larvae.

Reviewer #2:

1- I am very skeptical with the high variability shown in the binding assay in Supplementary Figure Some reasons are:

a - I would like to drag the attention of the authors to the crystal structures available for this strain in complex with HBGAs (<https://doi.org/10.1128/JVI.02968-14>) and with milk oligosaccharides (10.1128/JVI.03223-15). A simple superimposition of available crystal structures will show that carbohydrates show very similar (in some cases identical) interactions with the P domains, which strongly suggests similar affinities. As an example, please compare PDBs 4X0C (Lewis X) vs 4OP7 (B-tris).

b- In addition, dissociation constants KDs are available for a related strain GII.4 Saga 2006 (GenBank AB447457). Saga and Sydney share a 95% VP1 identity, with 100% conservation of amino acids located in the binding pocket. Comparison of crystal structures of Saga (4X06) and of Sydney (4OP7) in complex with B-tris show again identical interactions. NMR data available for Saga P dimers reveals very similar KDs for A- and B-tris.

c- Also, Lea and LeX possess almost identical 3D structural principles, which are in turn responsible for their recognition by hNoV (10.1016/j.coviro.2018.04.007). Why should they exhibit such a large difference in Abs?

In summary, these results need to be taken with care. In my opinion, the differences in Abs observed between HBGAs are likely due to variable multivalent presentation, rather than real affinity. That is, it is highly probably that the short and rigid acetyl-phenylenediamine link used by IsoSep AB to connect the carbohydrates with the HSA is influencing the outcome of the experiment. The authors should at least discuss it in the Supplementary Figure 1.

The authors agree with the comments of the reviewer. It's true that different binding assays give slightly different results. In this case the main differences with previous NMR data are that we used complete VLPS and conjugated glycans to HSA. So, in this case we have a polyvalent VLP as well as a polyvalent antigen (conjugated HSA), this configuration might reflect better what occurs "in vivo" with a complete virus and the HBGAs being expressed also in a polyvalent manner. It's also true that small changes in the sequence of the GII.4 variants produce also changes in their ability to recognize different HBGAs.

The results obtained by us show recognition of most assayed glycans except Le^a and GM1. The text has been changed to better reflect the obtained results (line 147).

2. Line 281: What about bile acids? They have been shown to interact in different ways with several hNoV strains. Does it make any sense in the biology of zebrafish larvae?

Studies have indeed shown for bile acids to have a positive effect of HuNoV replication, and for some strains even to be essential (e.g. GII.3). Zebrafish have a functional liver with biliary ductal network by 72 hours post fertilisation and have orthologues to many mammalian genes related to bile production. It is thus likely that the virus interacts in some way with bile acids/salts in the larval intestine. It is however hard to speculate whether an enhancing effect occurs in the zebrafish intestine as the specific bile composition and secretion, especially at early larval stages, is unknown. We did show previously that GII.3 strains are able to replicate in zebrafish albeit to a lower titre than other genotypes tried (GII.2, GII.4, GII.6) (Van Dycke et al, 2019).

3. - It is always helpful to add the GenBank accession number of the hNoV construct overexpressed, in this case VLPs. To be more precise, I would like to know if the VLPs used in this study and named as "GII.4 2012 Sydney" are in fact Hu/GII.4/Sydney/NSW0514/2012/AU, with GenBank accession number JX459908.1. Please add.

The VLP clone used here is accession number MN248513.1, which is the sequence of the GII.4 sample we deposited previously (Van Dycke et al, 2019) and which we used for the infection experiments in this manuscript.

4.- Please swap order of appearance in the text of figure 2B and Figure 2C. The order should be the one of appearance in the text. Also correct Fig. 2 if required.

We have corrected Figure 2 to chronologically fit to the text.

5-. Line 219: "harvested each day pi.." Is pi a typo?

Day pi has been clarified and rewritten to dpi (days post infection).

September 30, 2022

Prof. Joana Rocha-Pereira
Rega Institute for Medical Research, KU Leuven
Minderbroedersstraat 10
Leuven 3000
Belgium

Re: Spectrum03157-22R1 (The role of histo-blood group antigens and microbiota in human norovirus replication in zebrafish larvae)

Dear Prof. Joana Rocha-Pereira:

Your manuscript has been accepted, and I am forwarding it to the ASM Journals Department for publication. You will be notified when your proofs are ready to be viewed.

Sincerely,

Robert de Vries
Editor, Microbiology Spectrum
